# Effective high-throughput RT-qPCR screening for SARS-CoV-2 infections in children

Felix Dewald[1,2,33], Isabelle Suárez[2,3,4,33], Ronja Johnen [5], Jan Grossbach [5,6], Roberto Moran-Tovar [7], Gertrud Steger[1], Alexander Joachim[8], Gibran Horemheb Rubio [1,9], Mira Fries[3,10], Florian Behr[3,10], Joao Kley [3], Andreas Lingnau[11], Alina Kretschmer [3], Carina Gude[5], Guadelupe Baeza-Flores[12], David Laveaga del Valle[12], Alberto Roblero-Hernandez[12], Jesus Magana-Cerino[12], Adriana Torres Hernandez[13], Jesus Ruiz-Quinones [12], Konstantin Schega [10], Viktoria Linne[3], Lena Junker[3], Marie Wunsch[1], Eva Heger[1], Elena Knops[1], Veronica Di Cristanziano [1], Meike Meyer[8], Christoph Hünseler[8], Lutz T. Weber[8], Jan-Christoffer Lüers [14], Gustav Quade[15], Hilmar Wisplinghoff[16], Carsten Tiemann[17], Rainer Zotz[18,19], Hassan Jomaa[20], Arthur Pranada [21], Ileana Herzum[22], Paul Cullen[23], Franz-Josef Schmitz[24], Paul Philipsen[25], Georg Kirchner[26], Cornelius Knabbe[27], Martin Hellmich [28], Michael Buess[10], Anna Wolff[10], Annelene Kossow[10,29], Johannes Niessen[10], Sebastian Jeworutzki [30], Jörg-Peter Schräpler[30,31], Michael Lässig[7], Jörg Dötsch[8], Gerd Fätkenheuer[2,3], Rolf Kaiser[1,4], Andreas Beyer[5,6,32,33], Jan Rybniker[2,3,4,33] & Florian Klein [1,2,4,33✉]

Systematic SARS-CoV-2 testing is a valuable tool for infection control and surveillance. However, broad application of high sensitive RT-qPCR testing in children is often hampered due to unpleasant sample collection, limited RT-qPCR capacities and high costs. Here, we developed a high-throughput approach ('Lolli-Method') for SARS-CoV-2 detection in children, combining non-invasive sample collection with an RT-qPCR-pool testing strategy. SARS-CoV-2 infections were diagnosed with sensitivities of 100% and 93.9% when viral loads were >10$^6$ copies/ml and >10$^3$ copies/ml in corresponding Naso-/Oropharyngeal-swabs, respectively. For effective application of the Lolli-Method in schools and daycare facilities, SEIR-modeling indicated a preferred frequency of two tests per week. The developed test strategy was implemented in 3,700 schools and 698 daycare facilities in Germany, screening over 800,000 individuals twice per week. In a period of 3 months, 6,364 pool-RT-qPCRs tested positive (0.64%), ranging from 0.05% to 2.61% per week. Notably, infections correlated with local SARS-CoV-2 incidences and with a school social deprivation index. Moreover, in comparison with the alpha variant, statistical modeling revealed a 36.8% increase for multiple (≥2 children) infections per class following infections with the delta variant. We conclude that the Lolli-Method is a powerful tool for SARS-CoV-2 surveillance and can support infection control in schools and daycare facilities.

---

A full list of author affiliations appears at the end of the paper.

The clinical course of COVID-19 in children is generally mild[1,2]. However, severe courses, deaths, and post-acute COVID-19 syndrome have been described and pose a risk to children when exposed to SARS-CoV-2[3,4]. Moreover, viral loads measured in infected children can be as high as those in adults[5], which is consistent with the transmission of SARS-CoV-2 among children and from children to adults[6]. In order to control SARS-CoV-2 infections, schools have been closed worldwide, resulting in the loss of approximately 50% of all school lessons in 2020[7]. However, while school closures can reduce SARS-CoV-2 transmissions when imbedded in a general lock-down strategy, the negative impact on the development and health of children can be substantial and is manifested by e.g. higher rates of reduced emotional well-being, severe eating disorders, and overt psychiatric disease[8–10].

Early detection of SARS-CoV-2 infections can contribute to infection control[11]. In addition, SARS-CoV-2 surveillance in schools allows to determine the efficiency of non-pharmaceutical interventions (NPIs)[12]. Therefore, several test strategies for schools and daycare facilities have been developed. In these, samples were mostly obtained by self-sampling using rapid antigen detection tests (RADTs) or RT-qPCR analyses[13–16]. However, various challenges remain including reduced sensitivity of RADTs[17], acceptance of specimen collection by children[18], and limited RT-qPCR capacities[19]. Despite the recent authorization for SARS-CoV-2 vaccines in children aged 5–11 years[20], effective and sound test strategies can be critical to ensure infection control in open schools and daycare facilities. This is particularly important considering the dynamic situation of the SARS-CoV-2 pandemic in which new variants of concern (VOCs) emerge and spread[21]. Here, we developed a non-invasive sampling approach combined with high-throughput pooled RT-qPCR testing (Lolli-Method) followed by the design of a test concept for schools and daycare facilities. This test concept was successfully implemented as a SARS-CoV-2 screening program for over 800,000 children and demonstrated a precise monitoring and early detection of SARS-CoV-2 infections.

## Results

**Developing the Lolli-Method to screen for SARS-CoV-2 infections in children.** A widely applicable SARS-CoV-2 screening in children requires the combination of i.) an easy, safe, and non-invasive sampling method with ii.) a resource-saving, reliable and scalable SARS-CoV-2 testing method. To meet these requirements, we developed the Lolli-Method by which a regular swab is used for self-sampling, i.e. to be sucked on for 30 s (Lolli-swab), combined with a pooled RT-qPCR analysis. In order to determine the sensitivity of this method, we investigated 254 acutely infected individuals in a side-by-side sampling approach using Nasopharyngeal-/Oropharyngeal (Np-/Op) versus Lolli-swab. Lolli-swabs were collected under supervision and all samples were analyzed by RT-qPCR (Fig. 1a, Supplementary Fig. 1). By using the Lolli-Method, 95 out of 118 infected individuals were detected when sampled in the morning and 101 out of 153 when sampled during the day. Detected viral loads obtained by the Lolli-Method were lower (geometric mean $2.22 \times 10^3$ copies/ml) than viral loads measured in Np-/Op-swabs (geometric mean $6.36 \times 10^4$ copies/ml, $p < 0.0001$, Fig. 1b and c, Supplementary Data 1). However, while Lolli-swabs showed only 50% sensitivity in samples with corresponding viral loads of $<10^3$ copies/ml, diagnostic sensitivities of 91.4% and 100% were reached for matched Np-/Op-swabs with viral loads of $10^3–10^6$ and $>10^6$ copies/ml, respectively (Fig. 1d). Next, we determined the impact on the sensitivity by having food and liquid intake one hour before sampling (Fig. 1e, Supplementary Data 2), the use of

different swab-types (Fig. 1f, Supplementary Data 3) and the dilution effect by the pooling process (Fig. 1g, Supplementary Data 4). While different swab-types had no effect on sensitivity, breakfast one hour before sampling and pooling of up to 100 Lolli-swabs reduced the detected viral load by 2.2- and 3.3-fold, respectively. However, these differences did not result in a relevant reduction of overall sensitivity, detecting 56 out of 57 samples despite breakfast or pooling. Finally, the specificity of the Lolli-Method was found to be 100%, testing 55 healthy individuals individually with Np-/Op-and Lolli-swabs (Supplementary Data 5).

We concluded that the Lolli-Method is an easy and non-invasive method that is highly sensitive in detecting infected individuals with viral loads above $10^3$ copies/ml.

**High-throughput Lolli-Method screening concept in children.** Next, a screening concept for schools and daycare was developed. As part of this concept, Lolli-swabs of one class or group were obtained and pooled at sampling site, followed by RT-qPCR analysis. To this end, each child of a class received a Lolli-swab, performed self-sampling and placed it in a common 50 ml tube (Fig. 1h). Very young children or children with disabilities received assistance by their parents or teachers. The pooled samples were tested by SARS-CoV-2 RT-qPCR. In case the pool was tested negative, all children were assumed to be SARS-CoV-2 negative. In case the pool was tested positive, children of the positive pool were re-tested individually in order to identify the infected individuals (Fig. 1h).

To determine an optimal test frequency, the efficiency of a long-term SARS-CoV-2 screening was estimated using an SEIR model (Fig. 1i and j, Supplementary Table 1). Simulations of 8-week-test-periods were carried out for fully-connected populations of 20 individuals, performing ensemble averages of over $10^4$ runs. Simulations were performed for different basic reproduction values ($R_0$) of SARS-CoV-2 and different scenarios of SARS-CoV-2 prevalence in the general population (0.01% and 0.1%). The total number of infected individuals, the number of infections due to transmissions within the test population and the number of infections detected by the screening were calculated for different test frequencies (0, 1, 2, or 3 times per week) with a turn-around time of 1 day and mandatory quarantine of 14 days for all 20 individuals (See "Methods"). As a result, the proportion of prevented transmissions was 36–66%, 46–77%, and 53–82% for testing 1, 2, or 3 times per week, respectively (Fig. 1k). Taking logistics and limited RT-qPCR capacities into account, a test frequency of twice per week was considered most effective and was used for the subsequent implementation of a screening program.

**Implementing the Lolli-Method screening concept in schools and daycare.** The Lolli-Method screening concept was implemented as part of a governmental SARS-CoV-2 testing program in 3700 elementary schools and special needs schools in North Rhine-Westphalia, Germany, testing 742,771 students twice a week (Fig. 2a, Supplementary Table 2). Testing was mandatory for all students. Students had a median age of 8 years (IQR 2 years) with 354,125 (47.69%) being female and 388,646 (52.32%) being male. On average, 21.1 students were registered per class and 197.3 students per school (Fig. 2b). Sampling was conducted from calendar week 19 to 37 in 2021, which included 8 weeks before (calendar week 19–26) and 5 weeks after (calendar week 33–37) the summer holidays. During this period, the 7-day incidence in North Rhine-Westphalia ranged from 14.4 to 146.7 with a maximum in calendar week 34 (Fig. 2c). Notably, while at the beginning the variant of concern (VOC) alpha was

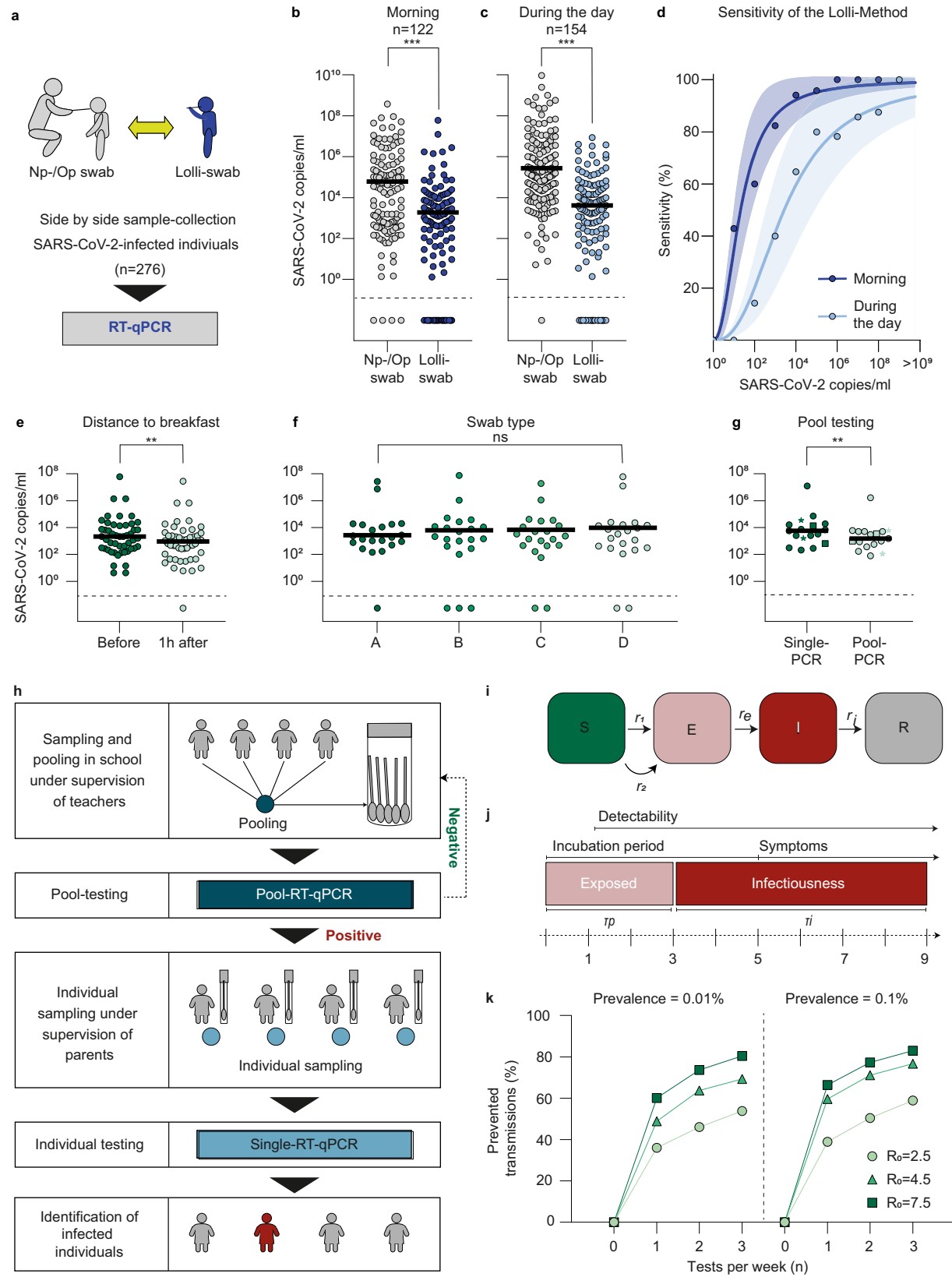

predominant, the delta variant accounted for the majority of cases starting with calendar week 26 (Fig. 2c).

For the 3700 schools that were located within an area of 34,098 km² sample transport as well as RT-qPCRs were performed by 12 diagnostic laboratories (Supplementary Fig. 2, Supplementary

Data 6). All RT-qPCR results were reported to a central database and data was checked for plausibility and invalid items were removed (Supplementary Fig. 3). In total, 1,110,033 RT-qPCRs were carried out (983,941 pool-and 126,092 single-RT-qPCRs). Average pool size was 10.2 and 16.7 Lolli-swabs/pool during calendar weeks

**Fig. 1 Lolli-Method for high-throughput SARS-CoV-2 screening in children. a** Experimental design used to validate the Lolli-Method. **b** Np-/Op-swabs and Lolli-swabs obtained in the morning plotted by viral load ($p < 0.0001$, two-tailed Wilcoxon signed rank test (WSR)). Horizontal lines represent mean viral loads. Asterisks represent $p$-values. **c** Np-/Op-swabs and Lolli-swabs obtained during the day plotted by viral load ($p < 0.0001$, two-tailed WSR). **d** The sensitivity of the Lolli-Method is stratified by viral load as fit curve (least squares method), indicated by both blue lines. 95% CI is indicated by colored area and time of sampling is indicated in corresponding colors. **e–g** Matched Lolli-Swabs plotted by viral loads obtained in the morning and 1 h after breakfast ($p = 0.021$, two-tailed WSR) (**e**), with four types of Lolli-swabs ($p = 0.72$, Friedmann test) (**f**), and for single-and pool-RT-qPCR, respectively ($p = 0.017$, two-tailed WSR) (**g**). **g** Dots represent pools of 18, squares of 49 and stars of 100 individuals. **h** Visualization of the screening concept. Samples are pooled in the classroom and tested in RT-qPCR. Individual RT-qPCRs are tested the next day and only in case of a positive pool. **i** SEIR-model to determine efficiency of the screening program. **j** Assumptions for the course of a SARS-CoV-2 infection. **k** Fractions of prevented transmissions stratified by test-frequency. Dots, triangles, and squares represent SARS-CoV-2 basic reproduction values of 2.5, 4.5, and 7.5.

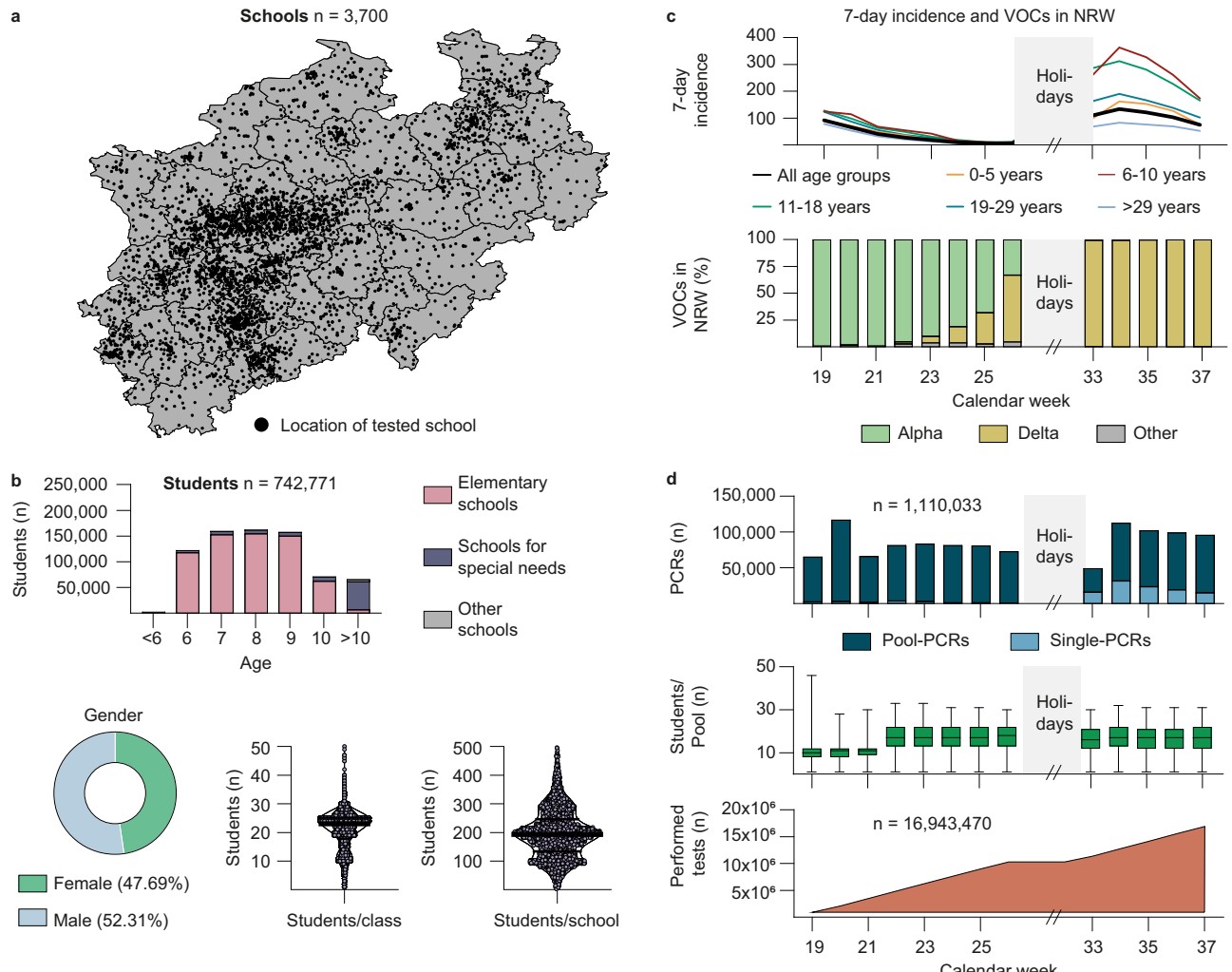

**Fig. 2 Implementing the Lolli-Method screening concept in schools and daycare facilities. a** Map of North Rhine–Westphalia. Black dots mark the location of each school. **b** Epidemiological characteristics of the tested students and characteristics of schools. Black horizontal lines in the violin plots represent median (*M*), first (*Q1*) and third (*Q3*) quartile (Students/class: $M = 23$, $Q1 = 18$, $Q3 = 25$; Students/schools: $M = 193$, $Q1 = 134$, $Q3 = 246$). **c** 7-day incidence of different age groups (top) and frequency of variants of concern according to data published by the Robert Koch Institute (bottom) stratified by calendar week. Summer holidays are marked in gray. **d** Number of performed RT-qPCRs (blue), average pool sizes (green) and total number of performed tests are stratified by calendar week. The horizontal lines in the green Box-Whisker-Plot indicate the medians, the lines at the top and at the bottom of the boxes indicate first and third quartiles and the error bars represent minimum and maximum pool sizes.

19–21 and 22–26/33–37, respectively, estimating an overall SARS-CoV-2 testing of 16,943,470 swabs within a time period of 13 weeks (Fig. 2d). Mean turn-around time for processing of pool-RT-qPCRs was 7.59 h and 9.14 h for single-RT-qPCRs (Supplementary Fig. 4). 96.2% of all pool-RT-qPCR results were communicated before 6:00 a.m. on the next day (Supplementary Data 6). In addition, the

Lolli-Method was applied to 698 daycare facilities in the city of Cologne, testing approximately 48,149 children within the age of 1–6 years and 13,577 staff members twice per week for a period of 6 months (Supplementary Fig. 4). Both in schools and daycare facilities, the program was well accepted and continued beyond the reported time period.

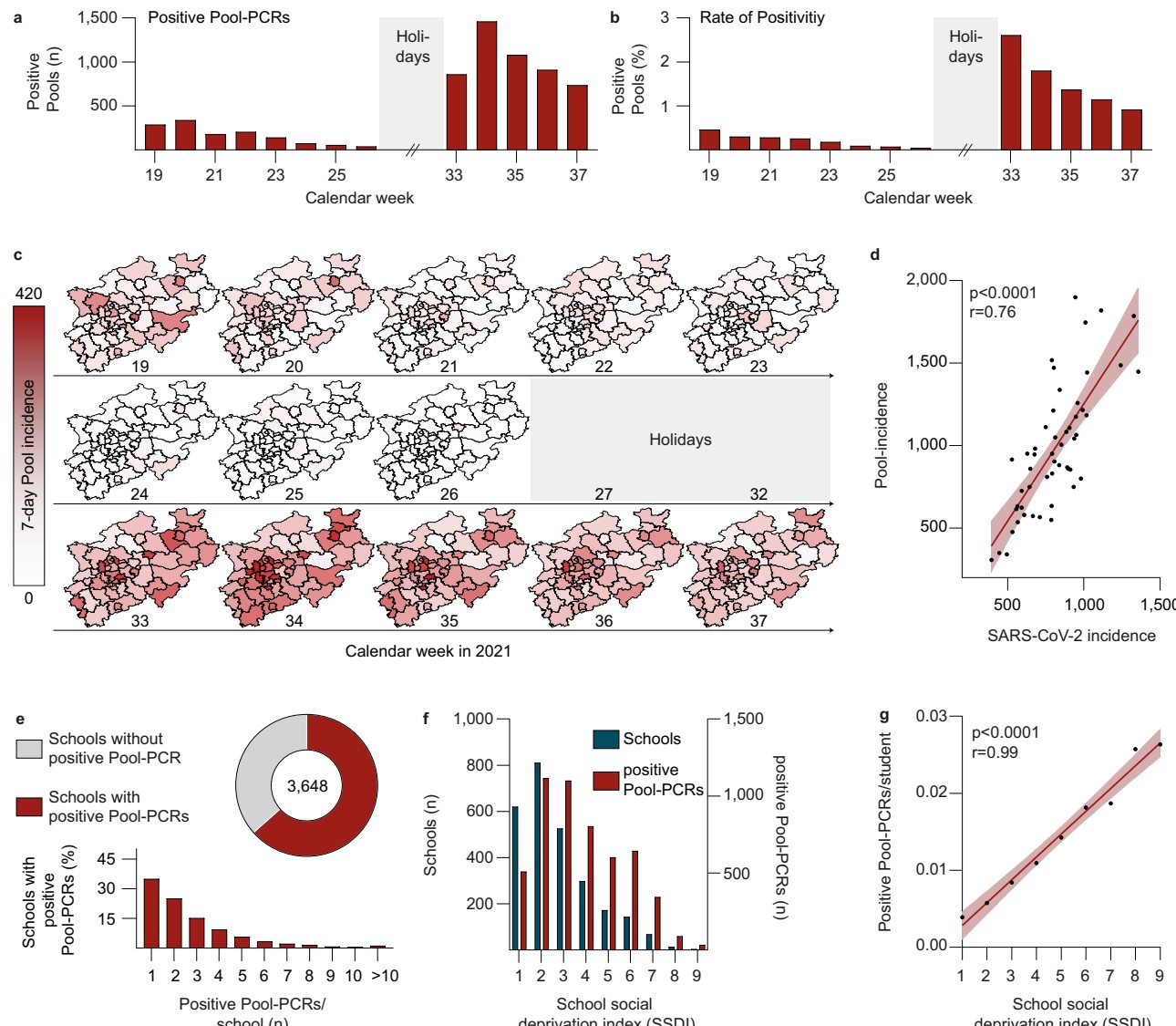

**Fig. 3 Monitoring SARS-CoV-2 infections in schools. a** The number of positive pool-RT-qPCRs is stratified by calendar week. **b** The rate of positivity of pool-RT-qPCRs is stratified by calendar week. **c** Maps of North Rhine-Westphalia depicting the 7-day pool-incidence per week and district. **d** Spearman correlation (two-tailed) between pool-incidence and SARS-CoV-2 incidence ($p < 0.0001$, $r = 0.76$). Each black dot represents one district of North Rhine-Westphalia. 95% CI is indicated by the bright red area. **e** Fraction of schools with at least one positive pool-RT-qPCR (pie-chart) and fraction of schools stratified by corresponding number of positive Pool-RT-qPCRs per school (bar chart). **f** Number of schools and positive pool-RT-qPCR stratified by school social deprivation index level. **g** Spearman correlation (two-tailed) between number of positive pool-RT-qPCRs/student and school social deprivation index level ($p < 0.0001$, $r = 0.99$). 95% CI is indicated by the bright red area.

We concluded that the Lolli-Method can be applied to educational settings including daycare facilities for high-throughput testing of SARS-CoV-2 infections.

**Monitoring SARS-CoV-2 infections in schools.** In total, 6364 of 983,941 pool-RT-qPCRs in schools tested positive (0.65%). 1316 pool-RT-qPCRs tested positive before (calendar week 19–26), while 5048 tested positive after the summer holidays (calendar week 33–37) (Fig. 3a). The rate of positivity of pool-RT-qPCRs was 0.46% in calendar week 19 and decreased continuously to 0.05% in calendar week 26. After the summer holidays, rate of positivity decreased from 2.61% in calendar week 33 to 0.92% in calendar week 37 (Fig. 3b). The number of infected individuals per positive pool-RT-qPCR was estimated to be 1.3 on average (Supplementary Fig. 6). In order to determine the false-negative rate of the implemented Lolli-Method, we investigated all reported index cases of children attending elementary schools in

the city of Cologne from calendar week 19 to 36 ($n = 653$, Supplementary Fig. 7). To this end, contact-tracing information was obtained by the local health authorities on 569 from 653 index cases (87.1%). When excluding index cases that were not tested by the Lolli-Method within 72 h before their positive test, detection rate of the Lolli-Method of confirmed index cases was 89.1% (Supplementary Fig. 7), indicating a reliable detection of SARS-CoV-2 infections in children. Furthermore, we confirmed the effect of the sample dilution by the pooling procedure described in the validation of the Lolli-Method (Fig. 1g) by comparing the Ct-values of the pool-RT-qPCRs and the corresponding single-RT-qPCRs (mean Ct-values 30.07 vs. 32.3, $p < 0.0001$, Wilcoxon-matched-pairs signed rank test; Supplementary Fig. 8).

SARS-CoV-2 7-day pool-incidence (= number of positive pool-RT-qPCRs/100,000 tested children in 7 days, see Methods, also for determination of number of samples per pool-RT-qPCR)

varied among districts from 0 to 416.2 (Fig. 3c) and correlated with SARS-CoV-2 incidence of the general population within a district ($r = 0.76$, $p < 0.0001$; Fig. 3d). Of 3700 participating schools, 3648 were tested before and after the summer holidays. Of those, 2315 (63.46%) schools were found to have at least one positive pool-RT-qPCR result, with numbers of positive pool-qPCRs per school ranging from 1 (22.2%), 2 (15.8%) and 3 (9.5%) to a maximum of 22 (0.03%, Fig. 3e).

Moreover, we investigated potential associations of SARS-CoV-2 infections in schools with grade levels, type of school, population density and socioeconomic status (SES) quantified using a school social deprivation index (SSDI)[22]. This index had been generated using a confirmatory factor analysis in which schools were assigned to social deprivation levels on a scale from 1 to 9 with 1 reflecting the highest SES and 9 the lowest. While no association between infections and grade levels or type of school was found, a moderate correlation for population density ($r = 0.56$, $p < 0.0001$) was detected. Moreover, the SSDI strongly correlated with the average number of positive pool-RT-qPCRs per student and per school ($r = 0.99$, $p < 0.0001$) (Fig. 3f, g).

We concluded that the Lolli-Method is capable of reliably detecting SARS-CoV-2 infections in schools and is a valuable tool to determine factors associated with SARS-CoV-2 infections in schools.

**High-throughput screening reveals differences in infection dynamics for SARS-CoV-2 variants in schools.** Based on molecular surveillance data published by the German public health institute (Robert Koch Institute) (Fig. 2c), we estimated the fraction of positive pool-RT-qPCRs assigned to the alpha variant (B.1.1.7) to be 92.9% before the summer holidays while 99.54% were assigned to the delta variant (B.1.167.2) after the summer holidays (Fig. 4a). Mean Cycle threshold (Ct)-values of positive pool-RT-qPCRs decreased significantly after the summer holidays, with an average Ct-value of 33.61 before (alpha variant) and 32.55 after (delta variant) the summer holidays ($p < 0.0001$, Mann-Whitney test). While the overall difference was small (1.06 Ct-values), pool-RT-qPCRs tested positive with high viral loads (Ct-value ≤25) were observed 3.1-fold more often for the delta variant compared to the alpha variant (Fig. 4b, c). Moreover, for viral loads detected with Ct-values ≤20, the difference between alpha and delta was even 7.6-fold.

In order to estimate a possible effect of the increase in pool-RT-qPCRs with low Ct-values on infection dynamics, the increase in positive pool-RT-qPCRs containing more than one infected child was statistically modeled, using data from calendar weeks 19–25 (alpha period) and calendar weeks 34–37 (delta period). The numbers of infected children per positive pool-RT-qPCRs expected by chance and without in-class transmissions (Null model) were estimated, while controlling for local incidence rates and the SSDI, and compared to the observed number (Methods and Supplementary Fig. 9). During alpha- and delta periods, numbers of positive pool-RT-qPCRs containing more than one infected child were 13 and 79, respectively, while 14.27 and 63.4 were expected based on the Null model (Fig. 4e). The ratio between observed and expected frequencies of pool-RT-qPCRs containing more than one infected child was 0.9 for the alpha period and 1.25 for the delta period (Fig. 4f). This amounted to an increase of 36.8% in positive pool-RT-qPCRs containing more than one infected child during the delta period.

We concluded that, during the delta period, more children with higher viral loads were present in schools and that parameters changing infection dynamics can be detected by applying the Lolli-Method in schools.

**Discussion**
During the pandemic, schools have been frequently closed to reduce SARS-CoV-2 transmissions[23]. However, closure of schools and daycare facilities can have a substantial impact on the development, physical and mental health of children[8–10,24]. Therefore, concepts are essential to support safe and open school settings. This is particularly important as new VOCs emerge that may substantially change infection dynamics.

Systematic testing can prevent transmissions in educational settings and gain insights of measures for infection control in children[14]. In addition, effective test strategies may allow to use NPIs more specifically and to reduce quarantine measures to keep school absence of children to a minimum[14]. Effective screening strategies require an easy and non-invasive sample collection, high sensitivity assays for early detection of infections, and high-throughput application[25]. As one SARS-CoV-2 test strategy, RADTs have been used[26]. While RADTs have the advantage of providing immediate test results, disadvantages include limited sensitivity[17], variation in specimen quality[27], and limited feasibility of self-sampling by young children. Finally, a high acceptance was observed for sample collection based on the Lolli-Method as demonstrated in a previous study[28].

RT-qPCR-based approaches for SARS-CoV-2 screenings in schools have been described[15,29,30]. In these studies, different specimens, such as buccal and anal swabs as well as gargling solutions and saliva samples were obtained. Some of these sampling methods may cause difficulties, e.g. gargling solutions may increase the risk of viral transmission during sampling because of aerosol generation. Moreover, strategies that depend on sample-pooling in the diagnostic laboratory require significantly more capacities in comparison to processing Lolli-swabs that have already been pooled in schools[31,32]. Considering limited resources of RT-qPCR-capacities, the Lolli-Method can be advantageous. As demonstrated in this study, less than 1.2 million RT-qPCRs were needed to investigate a total of 16.5 million swabs. However, high SARS-CoV-2 incidences yield larger numbers of positive pools associated with increasing numbers of follow-up single RT-qPCRs, which reduces the benefit of a pooling strategy and limits the application of pool-testing in high-incidence settings[33].

There is an urgent medical need to determine the role and effect of NPIs in educational settings, such as school closures[34], mandatory mask usage[35], and split-class lessons[36]. Notably, the described screening was sensitive enough to detect biological differences of the infection dynamics between the alpha and the delta variant[37]. Thus, the Lolli-Method may be further used to assess infection dynamics introduced by new variants[21] as well as determine the impact of measures taken in schools to prevent SARS-CoV-2 infections. Moreover, we could show a correlation between infection rates in schools and regional SARS-CoV-2 incidence which is in line with previous studies[16,38].

Limitations of our report include aspects of the data quality. These contain i) reporting of the viral load as non-standardized Ct-values and ii) incomplete reporting of pool sizes. However, there is a high level of consistency and comparability of Ct-values since RT-qPCRs were performed by the same laboratories during the course of the screening. In addition, due to the obligation for students to participate in the screening program, we were able to estimate pool sizes by extrapolation from reported data. One limitation of pool testing is that it is particularly suitable for low to medium SARS-CoV-2 incidences. For this reason, we consider it necessary to develop scalable modifications for the test concept for high-incidence phases (e.g. pool-size adjustment or additional use of RADTs). Finally, we do not provide real life effectiveness data of the Lolli-Method screening program in direct comparison with other screening programs (e.g. RADT settings). Therefore, further analyses are necessary to determine the effectiveness to

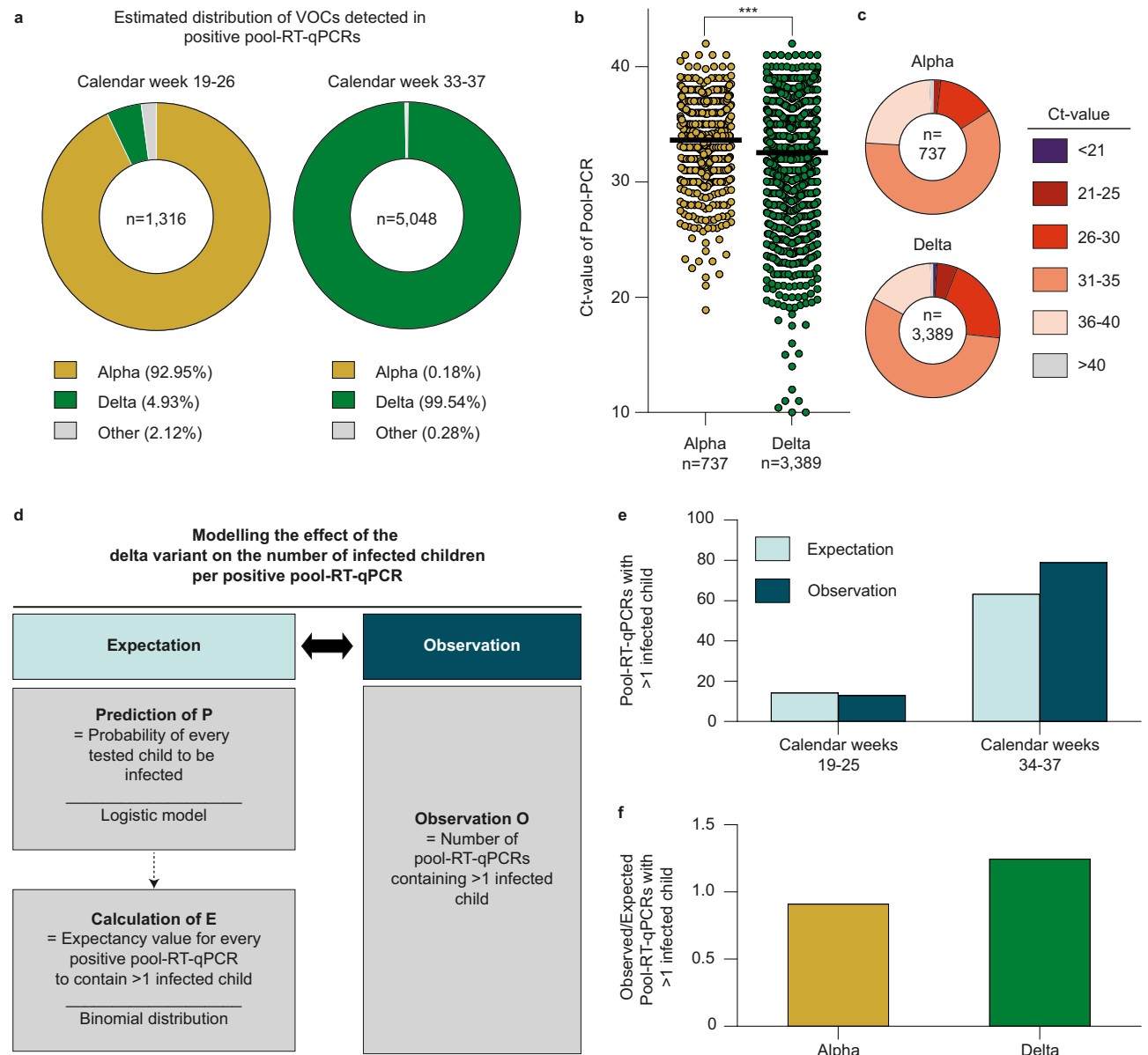

**Fig. 4 High-throughput screening reveals differences in infection dynamics for SARS-CoV-2 variants in schools. a** Estimated distribution of VOCs during calendar weeks 19–26 and 33–37. VOCs are indicated by corresponding colors. **b** All available Ct-values of positive pool-RT-qPCRs are plotted and stratified by estimated assignment to alpha ($n = 737$ pools) or delta ($n = 3,389$ pools) variant. Horizontal lines represent mean Ct-values, asterisks indicate the $p$-value ($p < 0.0001$, two-tailed Mann–Whitney test). **c** Categorization of Ct-values of positive Pool-RT-qPCRs assigned either alpha or delta variant in corresponding colors. **d** Expected and observed numbers of positive pool-RT-qPCRs containing >1 infected child were compared by statistical modeling. **e** The number of expected and observed positive pool-RT-qPCRs is stratified by time period. **f** The ratio of expected and observed numbers of positive pool-RT-qPCRs containing >1 infected child is stratified by SARS-CoV-2 variant.

reduce SARS-CoV-2 infections in children and the entire population using Lolli-testing in schools.

In summary, we developed, validated and implemented a non-invasive and sensitive technique for SARS-CoV-2 (self-)sampling that can be used for high-throughput application and screening. We consider this sampling method applicable to schools and daycare facilities providing a reliable tool for screening and surveillance of SARS-CoV-2 infections in children.

# Methods
## Ethical considerations
*Prospective validation of the Lolli-Method.* The prospective validation study was approved by the Institutional Review Board (IRB) of the Faculty of Medicine and University Hospital of Cologne, Cologne, Germany (number 20-1405) as well as by the IRB of the High Specialty Regional, Villahermosa, Mexico (number

0130144). The participants were either study patients of the University Hospital Cologne, Germany or of the test center of the High Specialty Regional Hospital, Villahermosa, Tabasco, Mexico. All participants gave their written informed consent before the start of the study.

*Retrospective analysis of the SARS-CoV-2 screening in schools.* The retrospective analysis of the SARS-CoV-2 screening in schools by the University Hospital of Cologne was engaged by the Ministry of Education and Schools and approved by the IRB of the Faculty of Medicine, University Hospital of Cologne, Germany (number 21-1358). Since 10th of May 2021, under the direction of the Ministry of Schools and Education and as part of the governmental SARS-CoV-2 screening program "Lolli-Test NRW", two Lolli-tests per week combined with a pooled RT-qPCR analysis were mandatory for all students at elementary schools and special needs schools in the state of North Rhine-Westphalia. Because testing was mandatory and in line with German law, no informed consent was required and obtained. 12 diagnostic laboratories were involved in processing the Lolli-swabs. These laboratories transmitted anonymized, de-identifiable data to a digital

database (Medeora Köln GmbH) for quality assurance purposes. Data were transmitted for pool-RT-qPCRs and single RT-qPCRs. For pool-RT-qPCRs, date of sampling, time of registration and result communication, the name of the school, the name of the class, the number of students per pool and the test result were transmitted. For single-RT-qPCRs, date of sampling, time of registration and result communication, the name of the school, the name of the class, age, gender and the test result were transmitted. From the digital database, data were transmitted to the University Hospital Cologne for retrospective analysis.

*Retrospective analysis of the SARS-CoV-2 screening in daycare facilities*. For retrospective analysis of the SARS-CoV-2 screening in daycare facilities, the University of Cologne was engaged by the Youth Welfare Office of the city of Cologne and approved by the IRB of the Faculty of Medicine, University Hospital of Cologne, Germany (number 21-1358). Since the 15th of March 2021, voluntary SARS-CoV-2 testing was offered to all daycare facilities in Cologne within the SARS-CoV-2 screening program "Kita Testung Köln (KiKo)" under the direction of the Youth Welfare Office of the city of Cologne. Of all participating children and staff members, two Lolli-Swabs per week were tested in a pooled RT-qPCR. One diagnostic laboratory was involved in processing the Lolli-swabs. This laboratory transmitted anonymized, de-identifiable data to the University Hospital of Cologne for retrospective analysis weekly. Data were transmitted for pool-RT-qPCRs and single-RT-qPCRs. For pool-RT-qPCRs, date of sampling, time of registration and result communication, the name of the daycare facility, the name of the group, the number of individuals per pool and the test result were transmitted. For single-RT-qPCRs, date of sampling, time of registration and result communication, the name of the daycare facility, the name of the group, age, gender and the test result were transmitted.

**Instructions for the SARS-CoV-2 screenings in schools and daycare facilities**. All staff, parents and children were instructed by either the Ministry of Education and Schools of North Rhine-Westphalia or the Youth Welfare Office of the city of Cologne. Written instructions in 12 different languages as well as instructional videos were used for training of all involved individuals. In addition, information and instructions for parents, children and staff were made available online (https://www.schulministerium.nrw/lolli-tests, https://www.kita-testung-koeln.de).

**Sample processing**
*Validation of the Lolli-Method*. To determine the sensitivity of the Lolli-Method, matched Lolli-swabs and Np/Op-swabs of acutely infected individuals were obtained. The participants were instructed to suck on a regular swab for 30 s. Very young children were supported by either their parents or a physician. Afterwards, a physician took an Np-/Op-swab.

To find out, whether the time of day at which the samples were taken had an impact on the sensitivity, the individuals were sampled either in the morning, one hour after breakfast or at any time of the day. The impact of the pooling process on the measured viral loads was determined by obtaining two Lolli-swabs from the same participant at the same time. One sample was tested in a pool-RT-qPCR with up to 17, 49, or 99 negative samples and the corresponding sample was tested in a single-RT-qPCR. To determine whether the detection rate depends on a particular swab type that is used for the sampling, the following four types of swabs were used for sample collection of Lolli-swabs: A) Oropharyngeal swab, Copan, catalog number: 801U059, B) Nasopharyngeal swab, Biocomma, catalog number: YVJ-TE4, C) Oropharyngeal swab, Biocomma, catalog number: YVJ-TE4, D) Dry swab, Sarstedt, catalog number: 1U059S01.

When tested in a single-RT-qPCR, each Lolli-swab was placed in a 2 ml tube pre-filled with 2 ml phosphate buffered saline (PBS), moved up and down and pressed against the bottom of the tube repetitively for 20 s. The Np/Op-swabs were vortexed in the viral transport media for 20 s. When tested in a pool-RT-qPCR, a Lolli-swab of one acutely infected individual was tested in a pool with 17, 49 or 99 Lolli-swabs of individuals not infected with SARS-CoV-2. A pool of Lolli-swabs was processed by placing the Lolli-swabs in one 50 ml centrifugation tube, adding 3 ml PBS and vortexing for 30 s. Of all samples, 1 ml each was used for SARS-CoV-2 detection.

*SARS-CoV-2 screening in schools and daycare facilities*. Lolli-swabs were used as described above for the sampling of the students in schools and children and the staff in daycare facilities. The staff was instructed for self-sampling and supervising of the sampling of the children. The samples of all participants of the same daycare group or school class were placed in one 50 ml centrifugation tube and transported to one of the 12 diagnostic laboratories. 3 ml PBS were pipetted in one centrifugation tube. The tube was vortexed for 30 s.

**SARS-CoV-2 detection and quantification**
*Validation of the Lolli-Method*. For SARS-CoV-2 detection, either COBAS 6800 (Roche Diagnostics) and Alinity m (Abbott) instruments equipped with their respective SARS-CoV-2 detection kits, or the Quantstudio 5 (Thermofisher) instrument, using the Quick-RNA Viral Kits (Zymo Research) for RNA isolation and GeneFinder™ COVID-19 Plus RealAmp was used.

For the comparison of cycle threshold (Ct) values measured by the different RT-qPCR equipments, Ct-values were translated into copies/ml. To this end, seven serial dilutions from a high titer SARS-CoV-2 sample were tested in all RT-qPCR equipments described above. With help of a regression model, standard curves for each equipment were generated. For the following conversion of device-specific Ct-values into copies/ml, two SARS-CoV-2 samples with a quantified RNA load from INSTAND (Society for the Promotion of Quality Assurance in Medical Laboratories, e.V., Düsseldorf, Germany; in cooperation with the Robert Koch Institute and the Institute of Virology, Charité, Berlin) were tested on every device and subsequently used for Ct-based absolute RNA quantification.

*SARS-CoV-2 screening in schools and daycare facilities*. For SARS-CoV-2 detection, the 12 laboratories reported to use different equipment which is listed in Supplementary Table 3. Viral load was reported as Ct-value.

**Adapting the SEIR-Model for a SARS-CoV-2 screening of children**. A compartmental epidemiological model was used to study the efficiency of a long-term SARS-CoV-2 screening based on the Lolli-Method. The model consists of a closed population of $N$ individuals and four possible states for each of them: Susceptible (S), Exposed (E), Infected (I), and recovered (R) (Fig. 1i). Those states were chosen based on the impact that a long exposed period has on the epidemiological dynamics and on testing-based non-pharmaceutical interventions[39].

Susceptible individuals get infected by a transmission within the population (internal infection rate) or by an exogenous transmission from outside the population (external infection rate), at rates $r_1$ and $r_2$, respectively. Overall, the total infection rate of susceptible individuals is given by

$$r = ar_1 + (1-a)r_2 \qquad (1)$$

where $a$ is the fraction of time that individuals interact within the test-population.

$$a = \frac{5\frac{\text{hours}}{\text{day}} \times 5\text{weekdays}}{12\frac{\text{hours}}{\text{day}} \times 7\text{day}} \approx 0.3 \qquad (2)$$

was chosen in order to approximate interactions within the test-population only occurring 5 h per day during weekdays (Mon-Fri) and assuming that on average there are in total 12 h per day of interaction in and outside the test-population. The internal infection rate is defined as

$$r_1 = \frac{\beta I}{N} \qquad (3)$$

and the external infection rate as

$$r_2 = \beta\pi \qquad (4)$$

where $I$ is the total number of infectious individuals in the population, $\pi$ is the global prevalence and $\beta$ is the infection-causing contact rate between individuals.

An infected individual that is in the exposed state moves into the infectious state at a constant rate of $r_e$:

$$r_e = \frac{1}{\tau_e} \qquad (5)$$

From the infectious state, an infected individual moves to the recovered state at a constant rate of $r_i$:

$$r_i = \frac{1}{\tau_i} \qquad (6)$$

These two last stochastic transitions follow homogeneous Poisson processes and therefore, exposed and infectious periods in the population follow exponential distributions with corresponding means $\tau_e$ and $\tau_i$.

**Implementing the sensitivity of the Lolli-Method in the extended SEIR-model**. The probability of a positive test result when testing an exposed or infected individual is $p_{\text{det}}$ given by

$$p_{\text{det}} = p_{\text{PCR}} \times S(\text{VL}) \qquad (7)$$

where $1 - p_{\text{PCR}}$ is the false-negative rate of RT-qPCR and $S(\text{VL})$ is a viral load dependent sensitivity function. To estimate $S(\text{VL})$, we fit the measured sensitivity data to a sigmoidal function

$$S(\text{VL}) = \frac{1}{1 + \left(\frac{\log_{10}\text{VL}}{\log_{10}\text{VL}_{50}}\right)^{-\delta}} \pm 1.96\sigma_S, \qquad (8)$$

where $\text{VL}_{50}$ and $\delta$ are fit parameters. $\text{VL}_{50}$ corresponds to the viral load at which $S(\text{VL}_{50}) = 0.5$ and $\delta$ quantifies the steepness of the sigmoidal function. We used the function *curve_fit* from the *scipy.optimize* library that implements a least squares method. The standard deviation for the fitted curve $\sigma_S$ was calculated with standard error propagation from the standard deviations of the fitted parameters,

$\sigma_{VL_{50}}$ and $\sigma_\delta$, as

$$\sigma_S = \frac{\left(\frac{\log_{10} VL}{\log_{10} VL_{50}}\right)^{-b}}{\left(\left(\frac{\log_{10} VL}{\log_{10} VL_{50}}\right)^{-b} + 1\right)^2} \sqrt{\left(\frac{\delta}{\log_{10} VL_{50}}\right)^2 \sigma_{VL_{50}}^2 + \left(\log \frac{\log_{10} VL}{\log_{10} VL_{50}}\right)^2 \sigma_\delta^2}, \quad (9)$$

such that $\pm 1.96\sigma_S$ represents 95% confidence level.

In order to determine the time dependence of the sensitivity of the Lolli-Method (days since infection), it was assumed that infected individuals would have a viral load of $10^6$ copies/ml three days after infection. We assume exponential growth for the viral load with constant rate $g$:

$$VL(t) = e^{gt} \quad (10)$$

Temporal dynamics of viral load and the associated sensitivity of the Lolli-Method are relevant because it was assumed that on average infectiousness would begin three days after infection (Fig. 1j). Thus, in this model, transmissions within the institutions can only take place when the infection occurred at least three days ago. Infected individuals would be infectious on average for 6 days. A summary of model inputs can be found in Supplementary Table 1[37,40–48].

The numerical dynamics consist of continuous-time and individual-based simulations, in which the transitions between states of each individual are stochastically determined using the Gillespie algorithm. Additionally, a testing scheme was implemented in which infected individuals were tested positive according to $p_{det}(t)$. Detected individuals were removed from the interacting population and the rest of the population was quarantined for the next 14 days one day after the detection. After this period, individuals could interact again. In this way, infected individuals that were not tested positive could transit to a recovered state without infecting other individuals in the test-population. We simulated testing protocols of 1 test per week (on Wednesday), 2 tests per week (Tuesday and Thursday) and 3 days per week (Monday, Wednesday and Friday).

**Calculation of 7-day pool incidence in schools**. Pool sizes were estimated by a linear model using official data by the Ministry of Schools and Education of North Rhine-Westphalia on class sizes and available reported pool sizes (Supplementary Fig. 10a). During the first three weeks of the screening, students were taught and tested in a split-class lesson model, while a full-class lesson model was the basis of lessons and testing for the rest of the screening. During split-class lessons, 50% of the students of one class attended lessons and were tested on Mondays and Wednesdays. The other 50% of the students attended lessons and were tested on Tuesdays and Thursdays. During full-class lessons, the whole class attended lessons daily and was tested either Mondays and Wednesdays or Tuesdays and Thursdays (Supplementary Fig. 10b). Pool sizes were reported by the tested schools (Supplementary Fig. 10c). Means of class sizes per school and means of reported pool sizes per school were mapped as part of a linear model with a forced Y-axis intercept of 0 (Supplementary Fig. 10d). Slopes during spit-class lessons were $m = 0.97$ and during full-class lessons $m = 0.97$ before and 0.96 after summer holidays. Thus, reported average pool sizes per school corresponded to approximately 97% and 96%, respectively, of the average class sizes per school. For this reason, average class sizes per school were used as an estimate of the average pool sizes per school for the estimation of number of children tested and the subsequently calculated 7-days-pool-incidence (pool-incidence = number of positive pool-RT-qPCRs/100.000 tested children in 7 days).

**School social deprivation index (SSDI)**. The level of social deprivation of schools is measured by a nine-level school social deprivation index (SSDI). The index is generated via a confirmatory factor analysis with four indicator variables[22]. The latent variable is divided into nine classes: Level 1 corresponds to a very low social deprivation; level 9 corresponds to a very high social deprivation. The index is based on several school-related indicators:

1. Child and youth poverty in the vicinity of an elementary school
2. Proportion of students with predominantly non-German family languages
3. Proportion of students who have moved to Germany from abroad
4. Proportion of students with special educational needs for learning, emotional and social development and language

The selection of the indicators is based on two criteria: First, to reflect socio-demographic variables relevant to school performance, and second, to avoid additional data collection and to use data that are uniformly available across the state.

With the exception of the indicator for child and youth poverty, all data come from the official school statistics of the state of North Rhine-Westphalia. A kernel-density estimate for the residential addresses of minors in unemployment/social-assistance-beneficiary households from the statistics of the Federal Employment Agency forms the indicator for child and youth poverty. It is a location statistic that shows the spatial density of minors in the vicinity of schools. The fourth indicator is included in the model as an interaction indicator (indicator of child and youth poverty * proportion of students with special educational needs). Therefore, a correlation between the interaction indicator and the indicator for children and youth poverty was allowed in the factor model.

The index shows good explanatory power for different learning outcomes when evaluated with the results of the centrally organized performance assessments VERA 3 (e.g. correlation with reading comprehension in German results in R^2 = 0.39).

The SSDI was formed for each school as a superordinate unit of several locations. These locations are referred to in the manuscript as „schools". The analysis of the correlation between SSDI and SARS-CoV-2 infections is based on the test data from these locations, but is carried out using the school as the superordinate unit.

**SARS-CoV-2 infection dynamics in schools**. Estimating the differences between SARS-CoV-2 variants in infection dynamics in schools was based on the notion that transmissions inside school classes (in-class transmissions) should lead to an excessive number of pool-RT-qPCRs containing more than one infected child. We implemented this notion by first estimating the expected number of pool-RT-qPCRs with more than one infected child assuming a Null model without in-class transmissions. We next compared the expected number of pool-RT-qPCRs with more than one infected child to the observed number of pool-RT-qPCRs with more than one infected child.

Positive pool-RT-qPCRs were filtered based on the following criteria to ensure only high-quality data is used for this analysis: i) only pool-RT-qPCRs that have a matching positive single-RT-qPCR, ii) number of following single-RT-qPCRs is within 20% of the pool size.

First, we calculated the probability of each tested child being infected under the Null model. To calculate the probability of each tested child being infected, we fitted a single logistic regression model for the entire testing period (calendar week 19–25, 34–37) using the local (district level) 7-day incidence of children aged between 6 and 10 years, the rate of positivity of pool-RT-qPCRs per district and per calendar week and the school social deprivation index (SSDI) as predictors (covariates),

$$p(P_{iw}) = p_{iw} = \beta_1 \cdot I_{iw} + \beta_2 \cdot PR_{iw} + \beta_3 \cdot SSDI_i + \beta_4 \cdot I_{iw} SSDI_i + \beta_5 \cdot PR_{iw} SSDI_i \quad (11)$$

where $P_{iw}$ is the event that a specific child in a positive pool-RT-qPCR in school $i$ and week $w$ is tested positive, $p_{iw}$ is the probability of being infected per child, $I_{iw}$ is the local 7-day incidence among children aged between 6 and 10 years, $PR_{iw}$ is the rate of positivity of pool-RT-qPCRs per district and per calendar week $w$, $SSDI_i$ is the school social deprivation index of school $i$ and $\beta_{1-5}$ are the regression coefficients.

After fitting the above regression coefficients, we calculated the probability of being infected per child $p_{iw}$ for each positive pool RT-qPCR. Each such pool contains at least one infected child by definition. We computed the probability of observing additional infected children assuming a binomial distribution,

$$P(X) = \frac{n}{k} \cdot p_{iw}^k \cdot (1 - p_{iw})^{n-k} \quad (12)$$

where $p_{iw}$ is the probability of each child being infected derived from the logistic regression model above, $n$ is the number of tested children in a pool-RT-qPCR and $k$ is the number of additional infected children. Using this binomial distribution, we computed the expected number of pool-RT-qPCRs with more than one infected child under the Null model (i.e. assuming no in-class transmissions). These values were compared to the observed number of pool-RT-qPCRs with more than one infected child in a given time period.

**Statistical analysis**. Geometric means were calculated for viral loads. Differences in viral loads were calculated with Wilcoxon-signed rank test (WSR) and Friedman test (FT). P-values <0.05 were considered significant. Sensitivity (positive percent agreement) and specificity (negative percent agreement) were calculated using RT-qPCR agreement. Differences in Ct-values during the screening program were calculated with Mann–Whitney test (MWT). Data analyses were done using the software GraphPad Prism (v.9), Microsoft Excel for Mac (v.14.7.3.) and R programing language (v. 3.5.2, stats package).

**Additional software**. Maps of North Rhine-Westphalia and Cologne were designed with the iMapU tool provided by iExcelU.

**Reporting summary**. Further information on research design is available in the Nature Research Reporting Summary linked to this article.

# Data availability
The data that support the findings of this study are provided with this paper. Source data are provided in Supplementary Data 1–6 and Supplementary Table 2.

# Code availability
Codes of the epidemiological simulations are available in the Github repository https://github.com/betoto008/lolli_testing[49] and in Supplementary Data 7. Codes of the

statistical modeling are available in the Github repository https://github.com/beyergroup/Lolli-Test-NRW.git[50] and in Supplementary Data 8.

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

## Acknowledgements

The authors thank all participants for the validation of the Lolli-Method and all children and staff in tested daycare facilities and schools. We thank Sascha Nickel and Anne Fries as well as all staff members of their daycare facilities for their impetus for the development of the Lolli-Method. We thank all members of the Institute of Virology, University Hospital Cologne. In particular, we thank Irina Fish, Ivonne Torre-Lage, Christina Hellriegel, and Esther Milz for processing SARS-CoV-2 samples and optimizing the logistics. We thank Carsten Tschirner (IExcelU) for supporting data vizualisation, Stephan Glaremin, Udo Neumann, Anja Kolb-Bastigkeit (Amt für Jugend, Arbeit und Soziales der Stadt Köln), Barbara Michaelis, Niklas Marizy (Gesundheitsamt Köln) for administrative support during implementation of the screening in Cologne. We thank Janna Seifried and Sindy Böttcher (Robert Koch Institute) for continuous support and

discussions, Norbert Schmeisser and Frederik Schmeisser (Medeora GmbH Köln) for supporting data collection, Kim Zun-Gon, Florian Korte and Hendrik Schöneborn (Boston Consulting Group) for organizational support, Christoph Gusovius from the Ministry for Schools and Education of the State of North Rhine-Westphalia, and Dietmar Klimas and Philip Graul (Labor ZotzKlimas, Düsseldorf, Germany) and Philipp Kirfel (Synlab) for supporting diagnostic analysis. Funding was provided by the German Ministry of Education and Research (BMBF) (registration number: 01KX2021) within Bundesweites Forschungsnetz „Angewandte Surveillance und Testung" (B-FAST) project of the „NaFoUniMedCovid19" consortium. Furthermore, funding was provided by the state of North Rhine-Westphalia, by the Bundesministerium für Bildung und Forschung (registration number: ZMI1-2521COR004) and by the German Research Foundation (DFG, CRC 1310). Finally, we acknowledge support for the Article Processing Charge from the DFG (German Research Foundation, 491454339).

## Author contributions

All authors contributed to sections relevant to their experience. F.D., F.K., R.K., I.S., G.F., A.B., and J.R. contributed to conception and design of the project. F.D. G.H.R., R.K., F.K., V.D.C., G.B.F., D.L.V., A.R.H., A.J., J.M.C., A.T.H., J.R.Q., K.S., M.M., J.C.L., C.H., L.T.W., E.H., E.K., and M.W. developed and validated the Lolli-Method. R.M.T. and M.L. developed the extension of the SEIR model. G.Q., H.W., C.T., R.Z., H.J., A.P., I.H., P.C., F.J.S., P.P., G.K., C.K. performed RT-qPCRs during the screening programs. M.B., A.W., A.K., J.N., and A.L. partnered as public health scientists. F.D., R.J., J.G, G.S., F.B., J.K., A.K., C.G., and M.F. performed data analysis. M.H., R.J., J.P.S. and J.G. performed statistical analysis. F.D., L.J., and V.L. managed the administrative framework. F.K., G.F., A.B., and J.D. supervised the project. F.D., R.M.T., J.G., R.J. S.J., and J.P.S. wrote the draft manuscript. All authors were involved in the editing of the final manuscript.

## Funding

## Competing interests

F.D., F.K., and R.K. hold EU-wide trademark protection for the terms "Lolli-Test" (018503959) and "Lolli-Methode" (018503958). All authors have no competing interests.

## Additional information

[1]Institute of Virology, Faculty of Medicine, University Hospital Cologne, University of Cologne, Cologne, Germany. [2]Center for Molecular Medicine Cologne, University of Cologne, Cologne, Germany. [3]Department I of Internal Medicine, Division of Infectious Diseases, Faculty of Medicine, University Hospital Cologne, University of Cologne, Cologne, Germany. [4]German Center for Infection Research, Partner Site Bonn-Cologne, Cologne, Germany. [5]CECAD Research center, University of Cologne, Cologne, Germany. [6]CECAD Cologne Excellence Cluster on Cellular Stress Responses in Aging Associated Diseases, University of Cologne, Cologne, Germany. [7]Institute for Biological Physics, University of Cologne, Cologne, Germany. [8]Department of Pediatrics, Faculty of Medicine, University Hospital Cologne, University of Cologne, Cologne, Germany. [9]Infectious Diseases Department, Instituto Nacional de Ciencias Médicas y Nutrición Salvador Zubiran, Mexico City, Mexico. [10]Health department of Cologne, Cologne, Germany. [11]Ministry of Schools and Education of North Rhine-Westphalia, Düsseldorf, Germany. [12]Centro de Investigación en Enfermedades Tropicales y Emergentes, Hospital Regional de Alta Especialidad, Dr. Juan Graham Casasús, Villahermosa, Mexico. [13]Bioclilab SA de CV, Villahermosa, Mexico. [14]Department of Otorhinolaryngology, Head and Neck Surgery, Faculty of Medicine, University of Cologne, Cologne, Germany. [15]MVZ Labor Dr. Quade & Kollegen GmbH, Cologne, Germany. [16]Labor Dr. Wisplinghoff, Cologne, Germany. [17]Labor Krone, Bad Salzuflen, Germany. [18]Institute for Laboratory Medicine ZotzKlimas, Düsseldorf, Germany. [19]Department of Haemostasis, Haemotherapy and Transfusion Medicine, Heinrich Heine University Medical Centre, Düsseldorf, Germany. [20]Synlab, Leverkusen, Germany. [21]Medizinisches Versorgungszentrum Dr. Eberhard & Partner, Dortmund, Germany. [22]Medizinische Laboratorien Düsseldorf, Düsseldorf, Germany. [23]MVZ Labor Münster, Münster, Germany. [24]Mühlenkreiskliniken, Minden, Germany. [25]Labor Mönchengladbach MVZ Dr. Stein und Kollegen, Mönchengladbach, Germany. [26]Eurofins Laborbetriebsgesellschaft Gelsenkirchen GmbH & Eurofins MVZ Medizinisches Labor Gelsenkirchen GmbH, Gelsenkirchen, Germany. [27]Heart- and Diabetes Center NRW, Medical Faculty, Ruhr-University Bochum, Institute for Laboratory and Transfusion Medicine, Bad Oeynhausen, Germany. [28]Institute of Medical Statistics and Computational Biology, Faculty of Medicine, University Hospital Cologne, University of Cologne, Cologne, Germany. [29]Institute for Hygiene, University Hospital Münster, Münster, Germany. [30]Faculty of Social Science, Ruhr-University Bochum, Bochum, Germany. [31]German Socio Economic Panel Study (SOEP), Berlin, Germany. [32]Institute for Genetics, Faculty of Mathematics and Natural Sciences, University of Cologne, Cologne, Germany. [33]These authors contributed equally: Felix Dewald, Isabelle Suárez, Andreas Beyer, Jan Rybniker, Florian Klein. ✉email: florian.klein@uk-koeln.de

