## [Peer Review File · Nature Communications]

Effective high-throughput RT-qPCR screening for SARS-CoV-2 infections in childrenReviewers' Comments:

Reviewer #1:

Remarks to the Author:

In this article, the authors present an effective method to surveil SARS-CoV-2 infections in children. The essence of the non-invasive specimen collection complemented with a RT-qPCR strategy. Furthermore, the authors implemented their testing strategy in a large number of schools and daycare facilities in Germany. The authors show correlations between school infections and incidence in the area and multiple infections were detected when compare delta vs. alpha variant. In summary, the authors make a case for their Lolli-method for surveillance among youngsters. The authors did a great job to their community by developing this assay. The paper is solid. Please answer the few questions below:

- a- Could the authors plot a graph in which the y-axis represents the Ct of pooled samples and x-axis shows the Ct of the single positive sample in the corresponding pool? What is the correlation of this curve? How does the plot change if more of 1 positive sample exists in the pool? Does the authors have any data of the false negative rate of the assay?
- b- How was the number of samples in the pool determined?
- c- Please provide information regarding the sequence of the primers used in the RT-qPCR assay as well as an in-silico specificity/cross-reactivity assessment.

Reviewer #2:

Remarks to the Author:

The paper "Effective high-throughput RT-qPCR screening for SARS-CoV-2 infections in children" by Dewald et al. reports interesting findings from a German program to screen schools and daycare facilities in Germany by means of "Lolli-Tests". The study provides a comprehensive overview of how the programme was implemented together with theoretical arguments for its design and might therefore provide highly interesting pointers to authorities that are interested in implementing similar programmes in the future.

Overall, the study is well-written and the literature is adequately described. In my review I focus on methodological and statistical issues that the authors might want to consider to further improve the manuscript.

Beyond that, after reading the paper it was not really clear to me what the central scientific takeaways from this study were, i.e. how it improved our understanding, setting aside the important aspect of showing that such programmes can be successfully implemented. For instance, going beyond the theoretical arguments, can one show that, e.g., incidence growth rates in school-aged children in regions where the programme have been implemented showed any differences compared to German regions without such a screening? The study mostly focussed on surveillance aspects in the discussion, but can the authors say anything regarding the real-world effectiveness of this programme in mitigating infections?

Methodological/statistical comments:

-) No error estimates for the test-specific sensitivities are given. Confidence intervals for proportions might be useful to communicate the uncertainty associated with these results, e.g., the Clopper-Pearson Interval or related intervals come to mind.
-) The "extended SIR" model appears to be quite similar to what large parts of the modelling community would refer to as an "SEIR model". Are there any conceptual differences between the "pre-infectious" and the "exposed" phase? Until reaching the methods section, I at least was confused why

screening strategies seem to be evaluated in an SIR framework without an exposed phase; future readers could be spared such a confusion...

-) From the model description it appears that no turnaround time of the tests are included? This is surprising, particularly given that the turnaround times were empirically evaluated and could therefore be plugged into the model in a well validated way. Previous modeling has also shown that the effectiveness of screening strategies crucially depends on turnaround times. Also, it appears that no contact tracing is implemented in the model and only identified cases are isolated. Does this align with the contact tracing policy that was implemented during the observation period in NRW? I can accept the exclusion of contact-tracing as a conservative modelling assumption (though this needs to be discussed), but the lack of (a discussion of?) turnaround times in the model appears to be a more substantial methodological issue.

-) The eq. shown on line 528 ($r = ar_1 + (1-a)r_2$) is an interesting modelling choice, but there are some minor consistency issues. From $a=5/7$ follows the interpretation that we are not mixing in-school and out-school transmission but schoolday and not-schoolday transmissions (the difference being that also on school days one can get infected outside of schools). The factor a can maybe better be taken from, e.g., PolyMod social mixing data or CoMix data to disentangle school-contacts from non-school contacts, whatever the authors have at their disposal. Furthermore, do I understand it correctly that even though some components of the model are individual-based, the equation discussed above hold for any given day? Or are there school days and non-school days in the model? On which weekdays were the tests implemented? These choices could also influence the results and should be described and/or discussed.

-) There appears to be a slight inconsistency across the different analyses that the paper combines in how correlations are evaluated. If I read it correctly, some results regarding infection risks and school deprivation index have been derived from a regression model whereas others from a bivariate correlation. Is this correct? Why not report all estimates from regression models that are structurally as similar as the available data allows? Or are Figures 3d and g already effect plots from the same regression model? The authors describe a potential confounding influence of population density, but I don't see an adjustment for that on line 646 in the regression model. This is confusing.

Point-by-point reply to comments of reviewers and of the Editor

Reviewer #1 (Remarks to the Author):

In this article, the authors present an effective method to surveille SARS-CoV-2 infections in children. The essence of the non-invasive specimen collection complemented with a RT-qPCR strategy. Furthermore, the authors implemented their testing strategy in a large number of schools and daycare facilities in Germany. The authors show correlations between school infections and incidence in the area and multiple infections were detected when compare delta vs. alpha variant. In summary, the authors make a case for their Lolli-method for surveillance among youngsters. The authors did a great job to their community by developing this assay. The paper is solid. Please answer the few questions below:

Response:

We thank the reviewer for the overall positive assessment of the work and our manuscript and for the helpful suggestions that we address below.

a.1- Could the authors plot a graph in which the y-axis represents the Ct of pooled samples and x-axis shows the Ct of the single positive sample in the corresponding pool? What is the correlation of this curve? How does the plot change if more of 1 positive sample exists in the pool?

Response:

We thank the reviewer for these interesting questions. We analyzed and plotted the Ct-values as proposed and the results can be found in **Extended data Fig. 8**. We differentiated between pool-RT-qPCRs that contained only one positive sample and those that contained more than one. Spearman coefficient was 0.4 for the correlation between Ct-values of pool- and single-RT-qPCRs when only Ct-values of pools with one positive sample were analyzed. Spearman coefficient was 0.31 when the average Ct-values of the single-RT-qPCRs from the same pool-RT-qPCR (pools containing more than one positive sample) were correlated with the Ct-values of the matched pool-RT-qPCRs (**Extended data Fig. 1a and b**). The rather weak correlation is probably based on separate days of sampling for pool- and single RTqPCRs (**Figure 1g**). Furthermore, we quantified the differences between Ct-values of pool-and single-RT-qPCRs (**Extended data Fig. 1a and b**). Detected differences are in line with the observed dilution effect seen in our evaluation of the PCR-pool-testing (**Fig. 1g**). Eventhough correlation was weaker when pools contained more than one positive sample, the difference in mean Ct-values between pool-and single-RT-qPCRs was lower. This observation can be explained by the assumption that pools with more than one positive sample contain more viral RNA, leading to a lower Ct-value. We have added the graphs to the manuscript as **Extended data Fig. 8** and commented in the revised manuscript in **Lines 160-164**.

a.2- Does the authors have any data of the false negative rate of the assay?

Response:

We thank the reviewer for this important question. We intensively investigated on the false-negative rate of the Lolli-Method by analyzing contact-tracing information obtained by the local health authorities. Briefly, the detection rate of the Lolli-Method of confirmed index-cases was 89.1%, suggesting a false-negative rate of 10.9%. We have explained the analysis more precisely in **lines 152-159** and **Extended data Fig. 7**.

b- How was the number of samples in the pool determined?

Response:

We thank the referee for this question that we consider very important. Therefore, we addressed the question in detail in the Methods section (**lines 616-636**) and in **Extended data Fig. 10**. In the revised manuscript we make this point more clearly and highlight this important aspect more precisely in the results part (**Lines 166-167**).

c- Please provide information regarding the sequence of the primers used in the RT-qPCR assay as well as an in-silico specificity/cross-reactivity assessment.

Response:

Since primer sequences are proprietary information by the manufactures of the used diagnostic instruments, we are unable to deliver these information. However, **Supplementary Table 1** lists the equipment that was used by the laboratories that participated in the screening. In-silico/cross reactivity assessments were performed as described by the manufacturers. In addition, all laboratories had to meet the quality requirements for virus diagnostics and had to pass the INSTAND e.V. quality controls which are distributed twice/four times a year in Germany. In our project, it was a prerequisite that the laboratories passed the requirements for SARS-CoV-2 RT-qPCR or SARS-CoV-2 TMA before they could take part in this project.

Reviewer #2 (Remarks to the Author):

The paper "Effective high-throughput RT-qPCR screening for SARS-CoV-2 infections in children" by Dewald et al. reports interesting findings from a German program to screen schools and daycare facilities in Germany by means of "Lolli-Tests". The study provides a comprehensive overview of how the programme was implemented together with theoretical arguments for its design and might therefore provide highly interesting pointers to authorities that are interested in implementing similar programmes in the future. Overall, the study is well-written and the literature is adequately described. In my review I focus on methodological and statistical issues that the authors might want to consider to further improve the manuscript.

Response:

We thank the reviewer for the appreciation of our study and for the constructive suggestions to improve our work. We address his/her suggestions below.

Beyond that, after reading the paper it was not really clear to me what the central scientific takeaways from this study where, i.e. how it improved our understanding, setting aside the important aspect of showing that such programmes can be successfully implemented. For instance, going beyond the theoretical arguments, can one show that, e.g., incidence growth rates in school-aged children in regions where the programme have been implemented showed any differences compared to German regions without such a screening? The study mostly focussed on surveillance aspects in the discussion, but can the authors say anything regarding the real-world effectiveness of this programme in mitigating infections?

Response:

We thank the reviewer for these important comments. We agree that assessing real-world effectiveness is very important. However, properly determining effectiveness is of great challenge given the various and constantly changing test strategies in other areas that could be used for comparison. While determining effectiveness is under investigation, we find that such an analysis is beyond the scope of the current study. Nevertheless, we included this topic in the discussion of the revised manuscript (**Lines 270-274**).

In terms of the scientific takeaway from this manuscript, we would like to respectfully disagree with the reviewer. From our point of view, various and novel scientific findings are provided with this study, including i.) a first comprehensive evaluation of a large-scale SARS-CoV-2 screening program, ii.) detected regional differences of infections in schools that are linked to SARS-CoV-2 incidences as well as to the socio-economic status of schools, and iii.) detection of biological differences between SARS-CoV-2 variants and their impact on infection dynamics in schools.

Methodological/statistical comments:

-) No error estimates for the test-specific sensitivities are given. Confidence intervals for proportions might be useful to communicate the uncertainty associated with these results, e.g., the Clopper-Pearson Interval or related intervals come to mind.

Response:

We thank the reviewer for this valuable comment. We agree and error estimates are plotted now in **Fig. 1 d**. A detailed description of the calculation of error estimates is given in the revised manuscript in **Lines 576-594**.

-) The "extended SIR" model appears to be quite similar to what large parts of the modelling community would refer to as an "SEIR model". Are there any conceptual differences between the "pre-infectious" and the "exposed" phase? Until reaching the methods section, I at least was confused why screening strategies seem to be evaluated in an SIR framework without an exposed phase; future readers could be spared such a confusion...

Response:

We agree that the originally used nomenclature might be confusing. Therefore, we changed the nomenclature in the revised manuscript to "SEIR-model".

-) From the model description it appears that no turnaround time of the tests are included? This is surprising, particularly given that the turnaround times were empirically evaluated and could therefore be plugged into the model in a well validated way. Previous modeling has also shown that the effectiveness of screening strategies crucially depends on turnaround times. Also, it appears that no contact tracing is implemented in the model and only identified cases are isolated. Does this align with the contact tracing policy that was implemented during the observation period in NRW? I can accept the exclusion of contact-tracing as a conservative modelling assumption (though this needs to be discussed), but the lack of (a discussion of?) turnaround times in the model appears to be a more substantial methodological issue.

Response:

We thank the reviewer for these important suggestions. We agree that the model can be improved by the implementation of turn-around times and contact-tracing policies. Therefore, we implemented a turn-around time of one day (**Lines 606-608**). Thus, it is assumed that the test result is communicated before the start of the next day in schools. This is in line with the observed turn-around times (**Extended data Fig 4**). Moreover, detected SARS-CoV-2-infected individuals were removed from the interacting population and the rest of the population was quarantined for the next 14 days one day

after the detection. This does align with contact-tracing in NRW during the time of modelling. We stated this more precisely in **Lines 606-608**.

-) The eq. shown on line 528 ($r=ar_1 + (1-a)r_2$) is an interesting modelling choice, but there are some minor consistency issues. From $a=5/7$ follows the interpretation that we are not mixing in-school and out-school transmission but schoolday and not-schoolday transmissions (the difference being that also on school days one can get infected outside of schools). The factor a can maybe better be taken from, e.g., PolyMod social mixing data or CoMix data to disentangle school-contacts from non-school contacts, whatever the authors have at their disposal. Furthermore, do I understand it correctly that even though some components of the model are individual-based, the equation discussed above hold for any given day? Or are there school days and non-school days in the model? On which weekdays were the tests implemented? These choices could also influence the results and should be described and/or discussed.

Response:

In the revised model, we assume that children attend school from Monday to Friday and from 8.00 a.m. to 01.00 p.m. (5 hours) as described in detail in **Lines 554-558**. Furthermore, we simulated testing protocols of 1 test per week (on Wednesday), 2 tests per week (Tuesday and Thursday) and 3 days per week (Monday, Wednesday and Friday) as described in **Lines 610-612**.

-) There appears to be a slight inconsistency across the different analyses that the paper combines in how correlations are evaluated. If I read it correctly, some results regarding infection risks and school deprivation index have been derived from a regression model whereas others from a bivariate correlation. Is this correct? Why not report all estimates from regression models that are structurally as similar as the available data allows? Or are Figures 3d and g already effect plots from the same regression model?

Response:

We thank the reviewer for this comment. The reviewer is correct in that the p-values in **Figures 3d** and **g** are based on bivariate correlation analysis while transmissions within the schools were modeled in a multivariate approach using SSDI and incidence as predictors. The individual analyses in **Figures 3d** and **3g** show that the district-level incidence and the SSDI individually correlate with the school incidence. However, the SSDI clearly correlates more strongly with the incidence than the school location (district). In our analysis, we have correlated the average school incidence per SSDI with the SSDI. Thus, we are averaging over many schools across districts. Therefore, it is not possible to include the district's incidence as another covariate in a linear model. In order to do so, one would have to model incidences at the individual school level (as opposed to averaging across schools), which would be much noisier. Notably, even at high regional incidences most schools had zero positive pools on a given day. However, when modeling the school positive rate per calendar week with the SSDI and district-level incidence in a logistic model, the coefficients are 0.22 and 0.015 (both p values <

2e-16) for SSDI and district-incidence, respectively, showing that both variables are meaningful predictors of the positive rate.

The authors describe a potential confounding influence of population density, but I don't see an adjustment for that on line 646 in the regression model. This is confusing.

Response:

We thank the referee for this question. This work observes a relationship between a broad sociological classification of schools (SSDI) and pool incidence. We do not attempt to differentiate between the possible causes of this increase in infections (e.g. language barriers, population density or access to masks etc.). As such analysis is highly complex and requires extensive additional data, we consider it to be out of scope for the present work.

Reviewers' Comments:

Reviewer #1:

Remarks to the Author:

The authors have properly addressed this reviewer's comments and further support its publication.

Reviewer #2:

Remarks to the Author:

I thank the authors for being so responsive and clarifying the methodological issues raised in the previous round. The authors resolved these issues comprehensively and I have no further major comments.

I spotted the following typos:

-) line 164 should say "Wilcoxon"

-) lines 546 and 572 still state "pre-infectious" instead of "exposed"

Point-by-point reply to comments of reviewers and of the Editor (25rd of april, 2022)

Reviewer #2 (Remarks to the Author):

I thank the authors for being so responsive and clarifying the methodological issues raised in the previous round. The authors resolved these issues comprehensively and I have no further major comments.

I spotted the following typos:

-) line 164 should say "Wilcoxon"

-) lines 546 and 572 still state "pre-infectious" instead of "exposed"

Response:

We thank the reviewer for the notice and we corrected both typos.